# Overexpression of Connexin 40 in the Vascular Endothelial Cells of Placenta with Acute Chorioamnionitis

**DOI:** 10.3390/diagnostics14080811

**Published:** 2024-04-12

**Authors:** Jia Yee Tan, Hannah Xin Yi Yeoh, Wai Kit Chia, Jonathan Wei De Tan, Azimatun Noor Aizuddin, Wirda Indah Farouk, Nurwardah Alfian, Yin Ping Wong, Geok Chin Tan

**Affiliations:** 1Department of Pathology, Faculty of Medicine, Universiti Kebangsaan Malaysia, Jalan Yaacob Latif, Bandar Tun Razak, Kuala Lumpur 56000, Malaysia; jiayee728@gmail.com (J.Y.T.); hannahyeohxy@gmail.com (H.X.Y.Y.); wk_chia@ppukm.ukm.edu.my (W.K.C.); tanweide@gmail.com (J.W.D.T.); indahfarouk@ppukm.ukm.edu.my (W.I.F.); nurwardah@ppukm.ukm.edu.my (N.A.); 2Department of Community Health, Faculty of Medicine, Universiti Kebangsaan Malaysia, Jalan Yaacob Latif, Bandar Tun Razak, Kuala Lumpur 56000, Malaysia; azimatunnoor@ppukm.ukm.edu.my

**Keywords:** chorioamnionitis, connexin, endothelial cells, placenta, gap junction

## Abstract

Background: Connexins (Cx) 43 and 40 play a role in leukocytes recruitment in acute inflammation. They are expressed in the endothelial cells. They are also found in the placenta and involved in the placenta development. Acute chorioamnionitis is associated with an increased risk of adverse perinatal outcomes. The aim of this study was to determine the expressions of Cx43 and Cx40 in the placenta of mothers with acute chorioamnionitis, and to correlate their association with the severity of chorioamnionitis and adverse perinatal outcomes. Methods: This study comprised a total of 81 cases, consisting of 39 placenta samples of mothers with acute chorioamnionitis and 42 non-acute chorioamnionitis controls. Cx43 and Cx40 immunohistochemistry were performed on all cases and their expressions were evaluated on cytotrophoblasts, syncytiotrophoblasts, chorionic villi endothelial cells, stem villi endothelial cells, maternal endothelial cells and decidua of the placenta. Results: Primigravida has a significantly higher risk of developing acute chorioamnionitis (*p* < 0.001). Neonates of mothers with a higher stage of fetal inflammatory response was significantly associated with lung complications (*p* = 0.041) compared to neonates of mothers with a lower stage. The expression of Cx40 was significantly higher in fetal and maternal vascular endothelial cells in acute chorioamnionitis (*p* < 0.001 and *p* = 0.037, respectively) compared to controls. Notably, Cx43 was not expressed in most of the types of cells in the placenta, except for decidua. Both Cx43 and Cx40 expressions did not have correlation with the severity of acute chorioamnionitis and adverse perinatal outcomes. Conclusion: Cx40 was overexpressed in the fetal and maternal vascular endothelial cells in the placenta of mothers with acute chorioamnionitis, and it may have a role in the development of inflammation in placenta.

## 1. Introduction

The placenta provides physiological exchange between the mother and fetus [1]. Acute chorioamnionitis (ACA) is defined as neutrophilic infiltrates of the chorion and amnion. If the inflammation involves the umbilical cord, it is known as acute funisitis [1]. ACA does not occur before the amnion and chorion fuse at about 11 weeks of gestation, and it is relatively rare before 19 weeks, prior to the fusion of placental membranes with the lower uterine segment [2]. The prevalence of ACA is 3% to 5% at term pregnancies and about 94% at the 21 to 24 gestational weeks pregnancies [1]. There are two different types of staging in ACA, i.e., maternal and fetal inflammatory responses. It depends on the site of neutrophilic infiltration in the placenta. Maternal inflammatory response is defined as neutrophilic infiltration in the fetal membrane, while fetal inflammatory response is defined as neutrophilic infiltration in the chorionic blood vessels and/or umbilical cord.

Acute chorioamnionitis with a higher MIR and FIR stage is associated with higher risk of adverse perinatal outcomes [1], including maternal complications such as postpartum haemorrhage, wound infection and endomyometritis, and neonatal complications like prematurity, neonatal sepsis, pneumonia, bronchopulmonary dysplasia, necrotising enterocolitis, cerebral palsy, cognitive impairment, intraventricular haemorrhage, hearing loss, impaired cardiac contractility and retinopathy of prematurity [2,3,4]. 

Connexins (Cx) are gap junction proteins consisting of two hemichannels (connexons) from adjacent cells that allow the exchange of ions and molecules [5,6]. Cxs are named based on their molecular weights (for example, Cx43’s molecular weight is 43kD) and their subgroups, e.g., α (GJA) or β (GJB) [5,7,8]. Cxs are expressed in various type of cells in humans, including endothelial cells and vascular smooth muscle cells [9]. In the endothelium, their functions range from basic processes of vascular remodeling and angiogenesis to vascular permeability and interactions with leukocytes in the vessel wall. Of the 21 members of the Cx family, 4 are expressed in endothelial and/or smooth muscle cells of blood vessels, including Cx37, Cx40, Cx43 and Cx45 [9]. Cx37, Cx40 and Cx43 are the three most dominant connexins that are expressed in the endothelial cells [6], and these are detected in the endothelial cells of both umbilical cord arteries and veins [10]. 

Notably, Cx43 and Cx40 have been found to be expressed in the placenta of early pregnancy [11], Cx43 in the villous cytotrophoblast and syncytiotrophoblasts and Cx40 in the extravillous trophoblasts and endothelial cells of villous arterioles [12,13]. Studies have shown that Cx43 plays a role in the fusion of cytotrophoblast into syncytiotrophoblasts through cell–cell communication and in the arrest of cytotrophoblast proliferation through intracellular signaling [11]. On the other hand, Cx40 regulates the proliferation and differentiation of extravillous trophoblasts during the first week of pregnancy. A loss or reduction of Cx40 contributes to the differentiation of extravillous trophoblastic cells into the invasive extravillous trophoblast phenotype [12].

Upon exposure to inflammatory stimuli, the normally closed connexins open and form a non-selective membrane pore that allows the passing of molecules of up to 1 kDa in size. This opening causes an efflux of potassium ions and influx of sodium, chloride and calcium ions. This subsequently leads to an increase in the cell permeability, cell depolarization and cell lysis. In addition, the opening also causes the release of adenosine triphosphate, glutamate, aspartate and oxygen free radicals, which leads to tissue injury. Thus, dysregulated connexins may be injurious and are known as “pathological pores” [14]. At present, the role of Cx in the placenta of mothers with acute chroioamnionitis is still largely unknown. Our study aims to characterize the expression patterns of Cx43 and Cx40 in the placenta with and without chorioamnionitis, and to correlate their expression with the severity of chorioamnionitis and adverse perinatal outcomes.

## 2. Materials and Methods

### 2.1. Study Design

This was a cross-sectional study comprising a total of 81 placenta tissue samples (39 cases of pregnancy with acute chorioamnionitis and 42 non-chorioamnionitis as controls). The samples were recruited over a period of 5 years. The formalin-fixed paraffin-embedded (FFPE) tissue blocks of placenta samples were retrieved from the archives of hospital laboratory medical services. The clinicopathological information such as maternal and gestational age, antenatal history, clinical diagnosis, histological diagnosis and staging of acute chorioamnionitis, baby birthweight, Apgar score and perinatal/neonatal outcomes were obtained from the integrated laboratory medical system and computerized medical record system (C-Hets) of the hospital. Patients’ information remained anonymous and each subject was coded accordingly. Cases of no umbilical cord or membrane, placenta accreta spectrum, pregnancy with hypertensive disorders, pregnancy with diabetes mellitus, chronic renal disease, autoimmune disease or insufficient clinicopathological data were excluded from the study. This study was approved by our institutional research ethics committee (approval code: FF-2021-105).

The haematoxylin and eosin-stained (H&E) slides of the selected cases were retrieved from the laboratory archive. The slides were reviewed independently by two pathologists (JYT and GCT) to determine the histological stage of acute chorioamnionitis (maternal inflammatory response and fetal inflammatory response) based on the staging system described by Redline et al. [2]. Two histological slides consisting of 1) the umbilical cord with membrane and 2) a full thickness were selected from each case to perform the Cx43 and Cx40 immunohistochemical staining.

### 2.2. Connexins Immunohistochemistry 

Two selected FFPE tissue blocks were stained with anti-Connexin 43/GJA1 and anti-Connexin 40/GJA5 antibodies.

1. Rabbit monoclonal anti-Connexin 43/GJA1 antibody [EPR22955-101] (Code: ab235585, Abcam, Cambridge, UK), at a dilution of 1:500. Adrenal tissue was used as positive control tissue. 

2. Rabbit polyclonal anti-Connexin 40/GJA5 antibody (Code: ab213688, Abcam, Cambridge, UK), at a dilution of 1:200. Lung carcinoma tissue was used as positive control tissue.

Immunohistochemical staining was performed following the protocol from Diagnostic BioSystems PolyVue™ Plus Kit (Code No. PVP250D, Diagnostic BioSystems, Pleasanton, CA, USA). Primary antibodies were diluted to optimal concentration using the antibody diluent, Dako REALTM (Code No. S2022, Dako, Groschup, Denmark). Washing steps between each reagent were performed using the EnVisionTM FLEX Wash Buffer 20 × (Code No. K8007, Dako, Groschup, Denmark), diluted to a working solution with deionized water. The DAB-containing Substrate Working solution was prepared by diluting the 50x concentrated EnVisionTM FLEX DAB+ Chromogen with EnvisionTM FLEX TM Substrate Buffer (Code No. K8023, Dako, Groschup, Denmark).

Tissue blocks were sectioned at approximately 3 µm thickness and mounted on an adhesive glass slide: Epredia Polysine Adhesion slide (Product No: J2800AMNZ, Dreieich, Germany). The slides were left to be air-dried at room temperature overnight. The tissue slides were then incubated on a hot-plate at 60 °C for 1 h. An initial deparaffinization and pre-treatment step was performed in the Decloaking Chamber™ NxGen (Ref. No: DC2012-220V, Biocare Medical, Pacheco, CA, USA) using the Dako Target Retrieval Solution High pH (Code No. K8023, Dako, Groschup, Denmark) with temperature 110 °C for 30 min, followed by cooling at room temperature for 30 min and rinsing with running tap water for 3 min. The slides were subsequently incubated with Tissue Primer (Code No. PVP250D) for 10 min followed by washing, then incubated with Background Blocker (Code No. PVP250D) for 10 min without washing. Slides were then incubated with a primary antibody for 30 min at room temperature followed by washing. Sections were then incubated with PolyVue™ (Enhancer) (Code No. PVP250D, Diagnostic BioSystems, Pleasanton, CA 94588, USA) for 30 min followed by washing. The slides were then incubated with PolyVue™ (HRP) (Code No. PVP250D) for 30 min at room temperature and followed by washing. Sections were then incubated with DAB-containing Substrate Working Solution for 7 min and then rinsed with washing buffer. The slides were then counterstained with Hematoxylin 2 (REF 7231, ThermoScientific, Waltham, MA, USA) for three dips, followed by dehydration with increasing alcohol solutions (80%, 90%, 100% and 100%) and two-times Xylene. Finally, the slides were mounted using CoverSealTM-X xylene-based mounting medium (Cat. No.: FX2176, Cancer Diagnostics, Durham, NC, USA).

### 2.3. Evaluation of Immunohistochemical Staining Percentage and Intensity

The percentage and intensity of Cx43 and Cx40 stainings were evaluated using an Olympus microscope (BX40, Olympus, Tokyo, Japan). The site of positive staining at the membrane, cytoplasmic and/or nucleus was recorded. The percentage of positive staining cells (scored on a scale of 0–2) and staining intensity (scored of a scale of 0–3) were added to obtain a total score of 0–5 (Table 1). A total score of 0–2 was regarded as negative, while 3–5 was regarded as positive. Both Cx43 and Cx40 expressions were evaluated on the 11 types of placenta cells: amnion epithelial cells (AEC), umbilical cord vascular endothelial cells (UVEC), umbilical cord vascular smooth muscle cells (UVSMC), chorionic villous endothelial cells (CVEC), chorionic villous vascular smooth muscle cells (CVSMC), stem vessels vascular endothelial cells (SVEC), stem vessels vascular smooth muscle cells (SVSMC), cytotrophoblasts (CYT), syncytiotrophoblasts (SYNT), maternal vascular endothelial cells (MVEC) and decidual cells (DC).

### 2.4. Statistical Analysis

Data analysis was performed using Statistical Package Analysis for the Social Science (SPSS) software version 27 (IBM SPSS Statistics version 27, Armonk, NY, USA). The evaluation of clinicopathological characteristics between the histological acute chorioamnionitis group and non-chorioamnionitis control group, and associations between maternal and fetal inflammatory responses with various adverse perinatal outcomes, were compared using the Chi-square test or Fisher exact test. A *p*-value of <0.05 was considered statistically significant.

## 3. Results

### 3.1. Demographic Data

This study consisted of 39 cases of ACA and 42 cases of non-ACA. The maternal age ranged from 22 to 39 years old (mean: 30.6 years old) and most of them were <35 years old. About 10% of cases in the ACA group were in the advanced maternal age (≥35 years old), with 21.4% of cases in the non-ACA group. In both ACA and non-ACA groups, most cases were of Malay ethnic group (ACA 29/39, 74.4%; non-ACA 34/42, 81.0%), followed by Chinese (ACA 6/39, 15.4%; non-ACA 4/42, 9.5%), Indian (ACA 1, 2.6%; non-ACA 1, 2.4%) and other ethnicities (ACA 3, 7.7%; non-ACA 3, 7.1%). In ACA group, the majority were primigravida (24/39, 61.5%), which was significantly different from the non-ACA group (5/42, 11.9%) (*p* < 0.001). Overall, 8 of the 39 cases (20.5%) of ACA were associated with preterm deliveries, higher than the non-ACA group (5/42, 11.9%) (*p* = 0.370). Of note, seven cases (7/39, 17.9%) of histologically confirmed ACA were not detected clinically. Table 2 summarizes the demographic and clinicopathological data of the cases with and without ACA in this study.

### 3.2. The Association of Stage of Maternal and Fetal Inflammatory Response of Acute Chorioamnionitis with Adverse Perinatal and Neonatal Outcomes

Most of the ACA were MIR stage 2 (17/39, 43.6%), followed by Stage 3 (12/39, 30.8%) and Stage 1 (10/39, 25.6%). Meanwhile, for FIR, most cases were distributed equally between Stage 1 and Stage 2 (12/39, 30.8%), and only seven cases were in Stage 3 (7/39, 17.9%). Overall, 8 of the 39 cases of ACA (20.5%) did not have FIR. Twelve of the cases with ACA had adverse perinatal outcomes, of which nine cases had premature delivery (8/39, 20.5%) and four cases (4/39, 10.3%) had either miscarriage or intrauterine death. Seven of the neonates in the ACA cases (7/35, 20.0%) subsequently developed lung complication like pneumonia and respiratory distress syndrome, which was significantly higher in cases with a higher stage of FIR (6/35, 17.1%) compared to cases with a lower stage (1/35, 2.9%) (*p* = 0.041). None of the neonates of cases with MIR Stage 1 had lung complications, compared to 36.8% (7/19) of the higher-stage MIR cases with lung complications. However, the difference was not statistically significant (*p* = 0.153) (Table 3).

### 3.3. Connexin 43 and Connexin 40 Expressions in the Various Types of Cells in the Placenta of Mothers with and without Histological Acute Chorioamnionitis

Cx43 was expressed in the membrane of decidual cells for all cases (42/42, 100%) of non-ACA and 97.4% (38/39) of ACA cases. Only two cases of ACA (2/39, 5.1%) showed cytoplasmic Cx43 expression in the AEC. Cx43 was not expressed in all the other types of cells in the placenta in both ACA and non-ACA. Cx40 was consistently positive in the SYNT and DC but was negative in the SVSMC and CYT. Of note, we found that Cx40 expressions in SVEC and MVEC were significantly different between ACA and non-ACA groups (*p* < 0.001 and *p* = 0.037, respectively) (Figure 1 and Figure 2). The difference in Cx40 expressions on other types of placenta cells between ACA and non-ACA (i.e., AEC, UVEC, UVSMC, CVEC and CVSMC) were statistically not significant (values of 0.485, 0.617, 0.620, 1.000 and 0.081, respectively) (Table 4) (Appendix A).

### 3.4. The Association between Connexin 43 and Connexin 40 Expressions and the Severity of Acute Chorioamnionitis and Adverse Perinatal Outcomes

There were no statistically significant differences between the expressions of Cx43 and Cx40 expressions with higher or lower stages of MIR and FIR. The Cx43 expressions in AEC and DC between MIR Stage 1 and Stages 2/3 were almost similar (*p* = 1.0) (Figure 3). The *p* values for Cx40 expressions in AEC, UVEC, UVSMC, CVEC, CVSMC, SVEC and MVEC between MIR Stage 1 and Stages 2/3 were 1.000, 1.000, 0.400, 1.000, 0.653, 0.721 and 0.302, respectively. Likewise, there were no statistically significant differences between the Cx43 and Cx40 expressions with adverse perinatal and neonatal outcomes (lung complications, Apgar score ≤ 3, intrauterine death/miscarriage, and prematurity) (*p* > 0.05) (Table 5).

## 4. Discussion

Connexins or gap junctions are involved in the intercellular communication which allows the exchange of ions and molecules and in turn facilitates electrical signaling and vasomotor control [9]. In this study, we found that in the placenta, Cx43 was not expressed in almost all types of cells, except for decidual cells. In addition, Cx43 expression remained the same in the placenta with ACA. In contrast, Cx40 showed variable expression in the placental cells, and the expression was increased in fetal and maternal endothelial cells. Previous studies reported that both Cx43 and Cx40 were expressed in the umbilical cord vessels [9,10,13]. Sarieddine et al. (2009) described an upregulation of endothelial Cx43 in acute lung injury [15]. In an animal study, Cx43 was found to be unchanged in the rat aortic endothelium in acute inflammation [16]. The inconsistent findings suggest that Cx43 expression may be differentially regulated based on its location during inflammation. Winterhager et al. (2015) reported that Cx43 was expressed in the decidual cells and may play a role in decidual angiogenesis and placental implantation [17].

Cx43 is the most frequently studied endothelial-related connexin and it plays a key role in leukocytes recruitment during acute inflammation [18]. The importance of Cx43 in leukocytes recruitment was evidenced by the reduction in neutrophils recruitment when it was treated with gap junction inhibitory peptide [9]. It was further supported by an in vivo study, which showed a reduction of leukocytes transmigration in connexin-deficient mice [18]. The expressions of Cx in the endothelial cells can be influenced by inflammatory mediators such as tumor necrosis factor-α, lipopolysaccharide and peptidoglycan. Robertson et al. reported that the opening of Cx43-hemichannel was induced by peptidoglycan and resulted in the release of ATP, which in turn induced interleukin-6 and Toll-like receptor 2 expression and led to leukocyte recruitment [9,18]. Sarieddine et al. (2009) reported that Cx43 was markedly upregulated in the lungs of mice when stimulated by lipopolysaccharide to simulate acute lung injury [15].

In this study, the Cx43 was expressed mainly in the decidual cells only. However, in the Human Protein Atlas portal (https://www.proteinatlas.org/), when using a rabbit polyclonal antibody Cx43 or Gap junction protein alpha 1 (GJA1) (CAB010753, Sigma Aldrich, St. Louis, MO, USA), it was positively expressed in the cytotrophoblasts, endothelial cells and Hofbauer cells, but not detected in the decidual cells [19]. We used the rabbit monoclonal anti-Connexin 43/GJA1 antibody [EPR22955-101] (Code: ab235585, Abcam, Cambridge, UK). One study reported the use of fresh tissue might be important as the majority of connexins including Cx43 have a rapid turnover rate and short half-life of 1.5 to 5 h after transportation to the plasma membrane, such as the plasma membrane of endothelial cells and smooth muscle cells. Connexins were later internalized and underwent degradation, thereby reducing the expressions [20].

Thus far, the results of Cx40 expression have been inconsistent across several studies. One study demonstrated the downregulation of Cx40 during acute lung injury and led to an increase in adhesion and the transmigration of neutrophils [15,18]. However, another study using a mice model showed that inflammation in acute lung injury was significantly reduced when there was a loss of Cx40 [21]. Rignault et al. (2007) found an upregulation of Cx40 during sepsis in the mice aortic endothelium [22]. 

Hemichannels are formed by hexamers of Cx, which contain a central pore permitting the movement of ions and molecules up to 1.2 kDa due to their electrochemical gradient. Under physiological conditions, the hemichannels are mostly closed to prevent leakage of molecules such as ATP and amino acids. However, under pathological conditions such inflammation and mutation, hemichannel activity may be increased and this represents a leakage route for molecules from the cells to extracellular space through the plasma membrane. This uncontrolled leaky flux of molecules leads to cell malfunction and cell death [13]. Intriguingly, we observed an overexpression of Cx40 in the stem villi fetal vascular endothelial cells and maternal vascular endothelial cells in acute chorioamnionitis compared to non-acute chorioamnionitis. This suggests Cx40 may be modulating the vascular tone to determine the blood flow in these blood vessels. This in turn could increase the recruitment of neutrophils to the site of infection. Alternatively, the endothelial cells may already be uncontrollably leaky and malfunctioning. 

Thus far, various distinct histomorphological features and immunomarkers have been identified in the placenta of acute chorioamnionitis, including CXC-chemokine receptor 1 and toll-like receptors [23,24,25,26]. We found about one fifth of the neonates of mothers with ACA and FIR had lung complications such as pneumonia and respiratory distress syndrome. Galinsky et al. (2013) reported that chorioamnionitis was often associated with FIR, and the neonates with FIR syndrome may result in poor cardiac, pulmonary, brain and renal outcomes [3]. The poor pulmonary outcomes include an increased risk of bronchopulmonary dysplasia, pneumonia and persistent pulmonary hypertension of the newborn [3].

## 5. Conclusions

We found that neonates of mothers with ACA and FIR have an increased risk of developing lung complications (pneumonia and respiratory distress syndrome), compared to neonates of mothers without ACA. Notably, Cx40 was overexpressed in the fetal and maternal vascular endothelial cells in the placenta. This finding suggests Cx40 may play a role in the pathogenesis of acute chorioamnionitis. Further study on Cx40 in acute chorioamnionitis may shed light to its role in the development of inflammation in placenta.

## Figures and Tables

**Figure 1 diagnostics-14-00811-f001:**
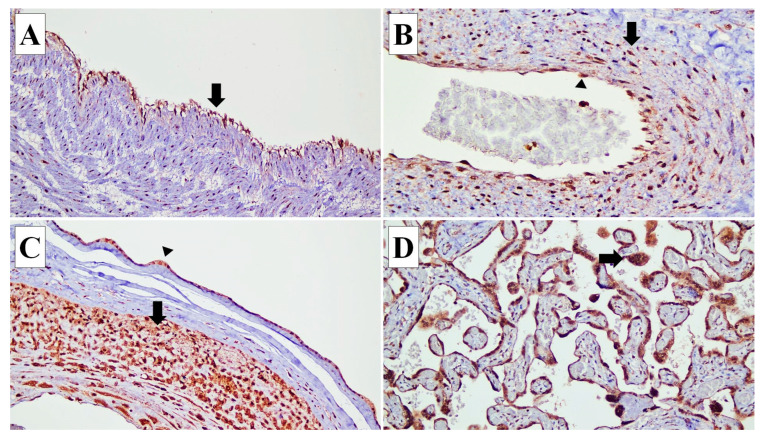
Connexin (Cx) 40 expressions in various types of cells in the placenta. (**A**) Cx40 expression is identified in the endothelial cells (arrow) of the umbilical cord blood vessels (×20). (**B**) In the blood vessels of stem villi, Cx40 is expressed in the endothelial cells (arrow head) and smooth muscle cells of the vascular wall (arrow) (×20). (**C**) In the fetal membrane, Cx40 expression is observed in the amnion epithelial cells (arrow head) and decidual capsularis cells (arrow) (×10). (**D**) Cx40 is expressed in the syncytiotrophoblasts (arrow) (×20).

**Figure 2 diagnostics-14-00811-f002:**
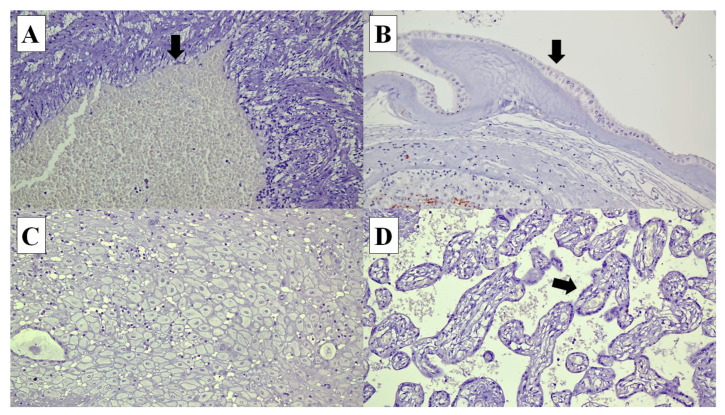
Connexin (Cx) 40 is not expressed in the following cells in the placenta: (**A**) umbilical cord vascular endothelial cells (arrow) (×20), (**B**) amnion epithelial cells (arrow) (×20), (**C**) decidual cells (×20) and (**D**) syncytiotrophoblasts in the chorionic villi (arrow) (×20).

**Figure 3 diagnostics-14-00811-f003:**
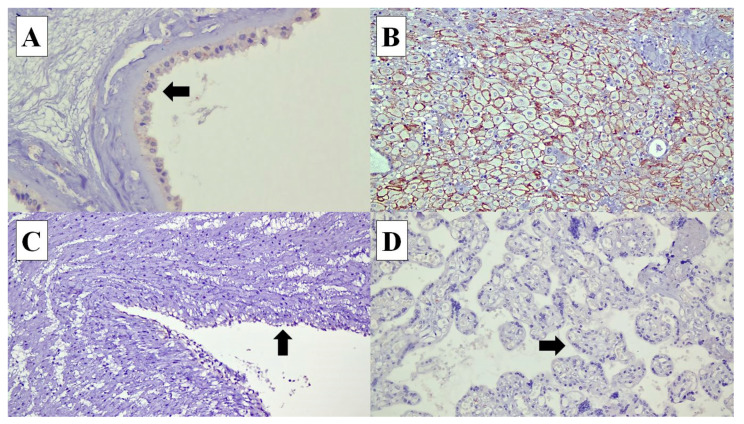
Connexin (Cx) 43 expressions in various types of cells in the placenta: (**A**) Cx40 expression is identified in the cytoplasm of the amnion epithelial cells (arrow) (×40). (**B**) In the decidual cells, Cx40 shows membranous staining pattern (×20). (**C**) Cx40 expression is not expressed in the umbilical cord vascular endothelial cells (arrow) (×20). (**D**) Syncytiotrophoblasts at the chorionic villi (arrow) (×20).

**Table 1 diagnostics-14-00811-t001:** Scoring of Cx43 and Cx40 immunohistochemistry.

Score	Percentage	Intensity
0	0–≤10%	No staining
1	>10%–<80%	Weak
2	≥80–100%	Moderate
3	-	Strong

**Table 2 diagnostics-14-00811-t002:** Demographic data of subjects with and without histological acute chorioamnionitis in this study.

Demographic	ACA		Non-ACA		*p*-Value
	Number of Cases (*n* = 39)		Number of Cases (*n* = 42)		
	No.	%	No.	%	
**Maternal Age (years)**					0.230
<35	35	90	33	79	
≥35	4	10	9	21	
**Ethnicity**					0.885
Malay	29	74	34	81	
Chinese	6	15	4	10	
Indian	1	3	1	2	
Others	3	8	3	7	
**Parity**					**<0.001 ***
Para 1	24	62	5	12	
Para 2–4	14	36	32	76	
≥Para 5	1	3	5	12	
**Gestational Age (weeks)**					0.370
<37	8	21	5	12	
≥37	31	80	37	88	
**Clinical suspected ACA**					
Yes	32	82	NA	NA	
No	7	18	NA	NA	

ACA—acute chorioamnionitis; NA—not applicable. * *p*-value < 0.05 was considered statistically significant.

**Table 3 diagnostics-14-00811-t003:** The association between stages of maternal and fetal inflammatory response (MIR and FIR) and adverse perinatal and neonatal outcomes.

	Apgar Score ≤ 3			IUD/Miscarriage			Prematurity			Lung Complications ^#^		
	Yes	No	*p*-Value	Yes	No	*p*-Value	Yes	No	*p*-Value	Yes	No	*p*-Value
**Severity of MIR**			1.000			1.000			0.653			0.153
Stage 1	0	9		1	9		1	9		0	9	
Stage 2/3	1	25		3	26		7	22		7	19	
**Severity of FIR**			0.486			1.000			1.000			**0.041 ***
Stage 0/1	0	18		2	18		4	16		1	17	
Stage 2/3	1	16		2	17		4	15		6	11	

IUD—intrauterine death, MIR—maternal inflammatory response, FIR—fetal inflammatory response. * *p*-value < 0.05 was considered statistically significant. ^#^ Lung complications—pneumonia and respiratory distress syndrome.

**Table 4 diagnostics-14-00811-t004:** Connexin 43 and Connexin 40 expressions in various types of cells in the placenta in acute chorioamnionitis and non-acute chorioamnionitis.

	Connexin 43			Connexin 40		
	Neg	Pos	*p*-Value	Neg	Pos	*p*-Value
**AEC**						
ACA	37	2 (C)	0.229	3	36 (C)	0.485
Non-ACA	42	0		6	36 (C)	
**UVEC**						
ACA	39	0	NA	1	38 (C)	0.617
Non-ACA	42	0		3	39 (C)	
**UVSMC**						
ACA	39	0	NA	30	9 (N)	0.620
Non-ACA	42	0		30	12 (N)	
**CVEC**						
ACA	39	0	NA	4	35 (C)	1.000
Non-ACA	42	0		5	37 (C)	
**CVSMC**						
ACA	39	0	NA	32	7 (N)	0.081
Non-ACA	42	0		40	2 (N)	
**SVEC**						
ACA	39	0	NA	27	14 (C)	**<0.001 ***
Non-ACA	42	0		40	2 (C)	
**SVSMC**						
ACA	39	0	NA	39	0	NA
Non-ACA	42	0		42	0	
**CYT**						
ACA	39	0	NA	39	0	NA
Non-ACA	42	0		42	0	
**SYNT**						
ACA	39	0	NA	0	39 (C)	NA
Non-ACA	42	0		0	42 (C)	
**MVEC**						
ACA	39	0	NA	5	34 (C)	**0.037 ***
Non-ACA	42	0		14	28 (C)	
**DC**						
ACA	1	38 (M)	0.481	0	39 (C)	NA
Non-ACA	0	42 (M)		0	42 (C)	

ACA—acute chorioamnionitis, Neg—negative, Pos—positive, AEC—amnion epithelial cells, UVEC—umbilical vascular endothelial cells, UVSMC—umbilical vascular smooth muscle cells, CVEC—chorionic villous vascular endothelial cells, CVSMC—chorionic villous vascular smooth muscle cells, SVEC—stem vascular endothelial cells, SVSMC—stem vascular smooth muscle cells, CYT—cytotrophoblasts, SYNT—syncytiotrophoblasts, MVEC—maternal vascular endothelial cells, DC—decidual cells, (C)—cytoplasmic staining, (M)—membranous staining, (N)—nuclear staining, NA—not applicable. * *p*-values < 0.05 were considered statistically significant.

**Table 5 diagnostics-14-00811-t005:** The association between Connexins 43 and 40 expressions and adverse perinatal outcomes in acute chorioamnionitis.

Connexins Expression	Apgar Score ≤ 3			IUD/Miscarriage			Prematurity			Lung Complications ^#^		
	Yes	No	*p*-Value	Yes	No	*p*-Value	Yes	No	*p*-Value	Yes	No	*p*-Value
**Cx43**												
AEC			1.000			1.000			1.000			1.000
Neg	1	32		4	33		8	29		7	26	
Pos	0	2		0	2		0	2		0	2	
DC			1.000			1.000			1.000			0.200
Neg	0	1		0	1		0	1		0	1	
Pos	1	33		4	34		8	30		28	6	
**Cx40**												
AEC			1.000			1.000			1.000			1.000
Neg	0	3		0	3		1	2		3	0	
Pos	1	31		4	32		7	29		25	7	
UVEC			1.000			1.000			0.205			1.000
Neg	0	1		0	1		1	0		0	1	
Pos	1	33		4	34		7	31		7	27	
UVSMC			0.200			0.223			0.065			1.000
Neg	0	28		2	38		4	26		6	22	
Pos	1	6		2	7		4	5		1	6	
CVEC			1.000			1.000			0.563			1.000
Neg	0	4		0	4		0	4		1	3	
Pos	1	30		4	31		4	27		6	25	
CVSMC			1.000			0.563			0.617			1.000
Neg	1	28		3	29		6	26		6	23	
Pos	0	6		1	6		2	5		1	5	
SVEC			0.371			1.000			0.424			0.220
Neg	0	22		3	22		4	21		6	16	
Pos	1	12		1	13		4	10		1	12	
MVEC			1.000			1.000			1.000			0.559
Neg	0	5		0	5		1	4		0	5	
Pos	1	29		4	30		7	27		7	23	

IUD—intrauterine death, Neg—negative, Pos—positive, AEC—amnion epithelial cells, UVEC—umbilical vascular endothelial cells, UVSMC—umbilical vascular smooth muscle cells, CVEC—chorionic villous vascular endothelial cells, CVSMC—chorionic villous vascular smooth muscle cells, SVEC—stem vascular endothelial cells, MVEC—maternal vascular endothelial cells, DC—decidual cells, ^#^ Lung complications—pneumonia and respiratory distress syndrome.

## Data Availability

Data are contained within the article and Appendix A.

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
