# Peer review of "Overexpression of Connexin 40 in the Vascular Endothelial Cells of Placenta with Acute Chorioamnionitis"

_diagnostics, 2024, doi:10.3390/diagnostics14080811_

Round 1

Reviewer 1 Report

Comments and Suggestions for Authors

In this article, Tan JY et al. study the expressions of Cx43 and Cx40 in the placenta of mothers with and without acute chorioamnionitis (ACA), and try to correlate their expression with the severity of chorioamnionitis and adverse perinatal outcomes. Although this is an interesting project, it is not a ready-to-publish work in its current state.

INTRODUCTION

- The first paragraph of Introduction section is quite confusing and the last two sentences are repetitive. Please, better define what acute chorioamnionitis (ACA) is and rephrase this paragraph.

M&M:

- The samples were recruited over a period of 5 years. Where these samples recruited with an informed consent approved by an Ethical Committee?

- FFPE tissue blocks of placenta samples were used. What parts of placental tissues were used?

- How is the percentage and intensity calculated? Is it calculated subjectively by the operator or does it refer to the number of positive cells?  Does the number of positive cells refer to the intensity? It is quite confusing, just as Table 1 is not well understood. How many fields of each sample are the authors analyzed?

RESULTS

- In the association of the stage of maternal and fetal inflammatory response of acute chorioamnionitis with adverse perinatal and neonatal outcomes, a study is lacking between how many cases of MIR and IRF occur simultaneously and whether these cases are associated with a severity in adverse and neonatal outcomes.

- Table 4: How many cells are positive and negative in each cell type analyzed? What is the intensity of the staining? The authors mention it in the M&M section, but do not describe it in the results.- Figure 1 is the expression of Connexin (Cx) 40 in various types of cells in the placenta. On which type of placenta is the image shown, ACA or non-ACA?. The authors should add a representative picture of all the cell types analyzed, positive and negative cases and ACA and Non-ACA samples.

- As well, the authors should also add a Figure with the expression of Connexin (Cx) 43 in all the cell types analyzed, positive and negative cases and ACA and Non-ACA samples.

- Please, make a table with the association between Connexin 43 and Connexin 40 expressions with the severity of acute 229 chorioamnionitis and adverse perinatal outcomes.

DISCUSSION

The authors should discuss their results first and make the discussion clearer.

CONCLUSION

It is difficult to understand the conclusion with the data presented.

Comments on the Quality of English Language

It is necessary to review the English with an English grammar expert because there are several sentences and paragraphs that are difficult to understand.

Reviewer 2 Report

Comments and Suggestions for Authors

Table 1 looks somewhat strange. A few lines above, the range of the score is given as 0-5. Furthermore, why is there nop ercentage given for intensity "strong"?

Statsitical Analysis: The name "Fisher" is Spelt incorrectly (as "Fischer").

Results: BEcause of the samll sample size, it makes little sense to specify percentages with one decimal place (Better rounding with no decimal places).

Section 3.4: p values should be presented. also in the case of non statitstically significant results. These results could also be due the rather samell sample sizes.

Round 2

Reviewer 1 Report

Comments and Suggestions for Authors

The authors have responded to all comments and suggestions and the article is ready for publication.